# Moxifloxacin Liposomes: Effect of Liposome Preparation Method on Physicochemical Properties and Antimicrobial Activity against *Staphylococcus epidermidis*

**DOI:** 10.3390/pharmaceutics14020370

**Published:** 2022-02-07

**Authors:** Evangelos Natsaridis, Foteini Gkartziou, Spyridon Mourtas, Marc C. A. Stuart, Fevronia Kolonitsiou, Pavlos Klepetsanis, Iris Spiliopoulou, Sophia G. Antimisiaris

**Affiliations:** 1Institute of Chemical Engineering, FORTH/ICES, Platani, 26504 Patras, Greece; natsaridis.e@gmail.com (E.N.); fotini_gartz@yahoo.gr (F.G.); klepe@upatras.gr (P.K.); 2Laboratory Pharmaceutical Technology, Department of Pharmacy, University of Patras, 26504 Patras, Greece; mourtas@upatras.gr; 3Electron Microscopy, Groningen Biomolecular Sciences and Biotechnology Institute, University of Groningen, Nijenborgh 7, 9747AG Groningen, The Netherlands; m.c.a.stuart@rug.nl; 4Department of Microbiology, School of Medicine, University of Patras, 26504 Patras, Greece; kolonits@upatras.gr (F.K.); spiliopl@upatras.gr (I.S.)

**Keywords:** moxifloxacin, antibiotic, liposome, active loading, drug retention, antimicrobial activity, *S. epidermidis*, cryo-TEM, release

## Abstract

The aim of this study was the development of optimal sustained-release moxifloxacin (MOX)-loaded liposomes as intraocular therapeutics of endophthalmitis. Two methods were compared for the preparation of MOX liposomes; the dehydration–rehydration (DRV) method and the active loading method (AL). Numerous lipid-membrane compositions were studied to determine the potential effect on MOX loading and retention in liposomes. MOX and phospholipid contents were measured by HPLC and a colorimetric assay for phospholipids, respectively. Vesicle size distribution and surface charge were measured by DLS, and morphology was evaluated by cryo-TEM. The AL method conferred liposomes with higher MOX encapsulation compared to the DRV method for all the lipid compositions used. Cryo-TEM showed that both liposome types had round vesicular structure and size around 100–150 nm, while a granular texture was evident in the entrapped aqueous compartments of most AL liposomes, but substantially less in DRV liposomes; X-ray diffraction analysis demonstrated slight crystallinity in AL liposomes, especially the ones with highest MOX encapsulation. AL liposomes retained MOX for significantly longer time periods compared to DRVs. Lipid composition did not affect MOX release from DRV liposomes but significantly altered drug loading/release in AL liposomes. Interestingly, AL liposomes demonstrated substantially higher antimicrobial potential towards *S. epidermidis* growth and biofilm susceptibility compared to corresponding DRV liposomes, indicating the importance of MOX retention in liposomes on their activity. In conclusion, the liposome preparation method/type determines the rate of MOX release from liposomes and modulates their antimicrobial potential, a finding that deserves further in vitro and in vivo exploitation.

## 1. Introduction

Moxifloxacin hydrochloride (MOX) is a fourth-generation fluoroquinolone with a dual antimicrobial mechanism of action [1]. While earlier fluoroquinolones had a limited spectrum of antimicrobial activity (mainly restricted to Gram-negative pathogens), numerous chemical modifications of the quinolone molecule resulted in increased activity, changes in pharmacokinetic characteristics, and decreased toxicity. In this context, MOX demonstrates enhanced activity against Gram-positive organisms and anaerobes and also reduces the resistance of more bacterial strains as compared to other fluoroquinolones [1,2,3,4]. The enhanced activity of MOX is related to the pathway of cell entry; it has been postulated that a mediator step involving a hydrophobic component that controls the extent/orientation of drug insertion into cell membranes finally improves the electrostatic interaction between the drug and the negatively charged cell membranes [5]. As a result of enhanced cell interaction, increased therapeutic properties of MOX as compared to other fluoroquinolones make it an ideal drug for the treatment of ocular infections, such as endophthalmitis; indeed, MOX has been proven as a safe prophylactic treatment following cataract surgery [6,7,8,9,10].

For treatment of localized ocular pathologies, the administration of drugs in the form of intraocular injections cannot be avoided due to the numerous biological barriers encountered between different eye segments [11]. Moreover, since intraocular injections have serious adverse reactions and side effects, the use of sustained-release drug formulations, such as liposomes, is preferred as a method to reduce the drug dosing frequency and/or enhance and prolong its therapeutic action [12,13,14,15,16,17,18,19,20].

Liposomes are nanosystems showing many advantages for antimicrobial drug-resistant infections, as recently reviewed [21]. MOX-loaded liposomes were recently demonstrated to have enhanced antitubercular activity in comparison with free drugs [22]. Herein we have investigated the development of MOX-loaded liposomes as a potentially improved treatment for ocular infections. For this purpose, two different methodologies were evaluated; the dehydration–rehydration vesicle (DRV) method, which is known to confer high drug loading in liposomes [23,24], and the active loading (AL) method [25,26], which was previously found to confer higher encapsulation of MOX in liposomes as compared to conventional passive loading such as thin-film hydration methods [26]. For the latter reason, the DRV method was selected as a passive loading method to compare with active loading. In addition to the effect of the liposome preparation method and lipid membrane composition on the vesicle physicochemical properties and the loading efficiency and release kinetics of MOX, a preliminary study to determine the antimicrobial activity of MOX-liposomes towards a reference biofilm producing bacterial strain of *Staphylococcus epidermidis* was conducted.

## 2. Materials and Methods

Egg phosphatidylocholine (PC), 1,2-Distearoyl-sn-glycerol-3-phosphatidyl-ethanolamine-N-[methoxy (polyethylene-glycol)-2000] (PEG), and Phosphatidyl-glycerine (PG) were purchased from Lipoid, Germany. Cholesterol (Chol) was purchased from Sigma-Aldrich (Darmstadt, Germany). Moxifloxacin was kindly supplied by COOPER (Athens, Greece). Agar was from Sigma-Aldrich (Darmstadt, Germany). All solvents used were of analytical or HPLC grade and purchased from Merck (Darmstadt, Germany). All other materials, such as salts used for buffer preparation, reagents for lipid concentration determination, etc., were of analytical grade and were purchased from Sigma–Aldrich (Darmstadt, Germany). Spectrapor^®^ dialysis membrane with an MWCO of 12–14 kDa was from Serva, Germany.

### 2.1. Preparation of MOX-Loaded Liposomes

Various lipid compositions were used for the formation of liposomes by both preparation methods. The lipid compositions tested, were: PC/Chol (2:1 and 1:1 mole/mole), PC/PG/Chol (9:1:5 and 9:1:10 mole/mole/mole), PC/PG/Chol (8:2:5 and 8:2:10 mole/mole/mole), PC/Chol/PEG (10:5:1.2 or 1.6 mole/mole/mole), and PC/PG/Chol/PEG (8:2:10:1.6 mole/mole/mole/mole). Details about liposome preparation using the two different methods applied are mentioned below. All liposomes were purified from non-encapsulated drug by size exclusion chromatography on Sephadex G-50 column (1 × 30), eluted with PBS, pH 7.40.

#### 2.1.1. DRV Method

For dehydration–rehydration vesicle (DRV) preparation, empty small unilamellar vesicles (SUV) were initially prepared as described in detail before [23,24]. For this, the lipids were dissolved in a chloroform/methanol (2:1 *v*/*v*) mixture and stored at −20 °C. For each liposome preparation, the appropriate amounts of each solution (depending on the lipid composition required) for a final lipid concentration of 20 mg/mL were placed in a 100 mL round-bottomed flask. Organic solvents were evaporated by the connection of the flask to a rotary evaporator until a thin film was formed. For complete removal of organic solvents, the film was flashed-dried with nitrogen for 2–3 min. The lipid film was hydrated with 1 mL of 10% PBS buffer pH 7.40 (5 mM PBS), heated at 40 °C. The resulting dispersions consisting of multilamellar vesicles (MLVs) were subsequently converted into SUVs by probe sonication with a tapered micro-tip (Vibra cell, Sonics and Materials, Suffolk, UK). In all cases, the initially turbid liposomal suspension was well clarified after sonication. Following sonication, the liposome suspensions (SUV) were left to stand for at least 1 h at 40 °C, in order to anneal any structural defects. Any Ti-fragment and/lipid aggregate contaminants were removed from SUV suspensions by centrifugation at 15,000× *g* for 20 min.

Then, 1 mL of the SUV suspension was mixed with 1 mL of a 10 mg/mL MOX solution (in distilled H_2_O), and the mixture was freeze-dried. With controlled rehydration of the dried materials as described previously [23,24], multilamellar dehydrated–rehydrated vesicles (DRV) were generated. Subsequent size reductions were carried out by sequential extrusion of the DRVs (10 times) through polycarbonate filters with pore diameter 0.4 µm and then 0.1 µm, fitted in a syringe-type extruder (Lipo-so-fast, Avestin, Ottawa, ON, Canada). Extrusion was used as a size-reduction method in order to prevent disruption of the DRV liposomes and leakage of the encapsulated drug.

#### 2.1.2. Active Loading Methods

For the preparation of MOX-liposomes by active loading, a previously reported active loading protocol was applied [25]. Empty SUV liposomes were initially prepared as described above, with the exception that ammonium sulfate (120 mM) was used for lipid film hydration. A pH gradient between interior/exterior of vesicles was then formed by replacing the external ammonium sulfate with PBS pH 7.40, by elution through a Sephadex G-50 (Merck) column (1 × 30 cm), with PBS. Active loading of MOX was then achieved by incubating liposomes (7 mg phospholipid) with MOX (3.5 mg) dissolved in PBS pH 7.40, in 1 mL total volume, at 40 °C for 1 h. The initial MOX/lipid *w*/*w* (D/L) ratio was thus 1:2, corresponding to a molar ratio equal to 0.947.

### 2.2. Physicochemical Characterization of MOX-Liposomes

#### 2.2.1. MOX Encapsulation Efficiency

Purified (from non-encapsulated MOX) liposomes were characterized for MOX encapsulation efficiency (%) calculated according to Equation (1):(1)$$\mathrm{Encapsulation}\;\mathrm{efficiency}\;\left(\%\right)=\frac{\frac{D}{L}\left(\mathrm{final}\right)\;\left(\frac{\mathrm{mol}}{\mathrm{mol}}\right)}{\frac{D}{L}\;\left(\mathrm{initial}\right)\;\left(\frac{\mathrm{mol}}{\mathrm{mol}}\right)}.\;100$$
where D is drug concentration and L is lipid concentration; initial means before purification and final after liposome purification.

Liposome lipid concentration was measured by the Stewart assay [27], a colorimetric method used for the quantification of phospholipids.

MOX concentration in liposomes was quantified by isocratic high-performance liquid chromatography (HPLC) using a Shimadzu 20A5 Gradient HPLC system coupled to aSPD-20A Prominence UV/VIS detector operating at 292 nm. A Luna^®^ 5 µm C18 (2) 100 Å, LC Column (250 × 4.6 mm), was used; the mobile phase was a mixture of acidified water (0.1% trifluoroacetic acid) and acetonitrile at 77:23 *v*/*v*. The column was eluted at a flow rate of 1 mL/min at 25 °C, and MOX was eluted at 12.3 min. Sample injection volume was 50 μL. Liposomes were analyzed after being lysed in methanol; one volume of sample was mixed with 10 volumes of methanol, and the mixture was agitated by vortex. A calibration curve in the range of 1–80 μg/mL was constructed by preparation of standard solutions of MOX in media with similar composition as the samples.

#### 2.2.2. Vesicle Physicochemical Properties

The particle size distribution (mean hydrodynamic diameter and polydispersity index) of MOX loaded liposomes dispersed at 0.4 mg/mL lipid, in phosphate-buffered saline (10 mM) with pH 7.40, was measured by dynamic light scattering (DLS) (Malvern Nano-Zs, Malvern Instruments, Malvern, Worcestershire, UK) at 25 °C and a 173° angle [28]. Each sample was measured 11 times in three independent measurements. The polydispersity index (PDI) was used as a measure of homogeneity of liposomal dispersions. Dispersions having a PDI of less than 0.200 or 0.250 are generally considered to have narrow size distribution. Zeta potential was measured in the same dispersions, at 25 °C, utilizing the Doppler electrophoresis technique, as recently reported [28].

The X-ray diffraction spectra of some liposome dispersions (after being precipitated by high-speed centrifugation), as well as solid MOX (for comparison), were recorded on a Bruker D8 Advance diffractometer. A zero background holder was used to obtain the spectra. The scanning range 2θ angle was from 5 to 40°, and the scanning speed was 1°/min. The spectra of the solid drug and some liposome samples were recorded in order to detect the potential presence of crystalline structures in the liposomes. Liposomes were centrifuged at 50,000× *g* rpm for 1 h, and a weighted quantity of the precipitate was used.

#### 2.2.3. Transmission Cryo-Electron Microscopy (Cryo-TEM)

The morphology of liposomes was studied by cryo-TEM. For this, MOX-loaded DRV and AL liposome dispersions were prepared as described above and re-dispersed in HEPES buffer (by ultrafiltration); finally, lipid concentration was adjusted at 5 mg/mL. A few microliters of each liposome dispersion were placed on holey carbon-coated copper grids (Quantifoil 3.5/1, Quantifoil Micro Tools, Jena, Germany). Grids with samples were vitrified in liquid ethane (Vitrobot, FEI, Eindhoven, The Netherlands) and transferred to an FEI Tecnai T20 cryo-electron microscope operating at 200 keV using a side entry cryo-stage (Gatan model 626). Images were recorded under low-dose conditions using a slow-scan CCD camera.

#### 2.2.4. Drug Release and Liposome Physical Stability Studies

A dialysis membrane method was used to follow the release kinetics of MOX from the different liposome types. Briefly, 0.5 mL of MOX liposomes (adjusted at a concentration of 1 mg/mL lipid) were placed in a dialysis bag constructed from dialysis tubing. Bags were immersed in 15 mL vials containing 15 mL PBS (pH 7.40) that were closed and placed in a shaking (50 rpm) incubator at 37 °C for up to 240 h. At predetermined time intervals, the entire medium was withdrawn and replaced with fresh; samples were assayed by HPLC for MOX concentration. Experiments were carried out in triplicate, and results are expressed as the mean amount released (%) ± standard deviation.

The physical stability of PC/Chol (1:1) MOX-loaded liposomes during storage at 4 °C for up to 120 d was determined by measuring the vesicle mean diameter, PDI, and ζ-potential of the different MOX-liposome types, as described above. At specified time intervals, the amount of drug retained in the vesicles was measured after separation from any drug that leaked out of vesicles using a Sephadex G-50 column (as described above).

### 2.3. Antimicrobial Activity (In Vitro)

#### 2.3.1. Bacterial Strains and Growth Conditions

The reference biofilm-positive strain, *Staphylococcus epidermidis* ATCC 35984, was grown aerobically in Tryptic soy broth (TSB, Oxoid CM0129, Oxoid Ltd., Wade Road, Basingstoke, Hants, RG24 8PW, Basingstoke, UK) and on Tryptic soy agar (TSA, Oxoid) plates at 37 °C overnight.

#### 2.3.2. Minimum Inhibitory Concentration (MIC) Determination

The minimum inhibitory concentration of free or liposomal MOX was determined by the broth microdilution method [29,30,31,32]. The bacterial strain was inoculated onto TSA plates and incubated overnight; a bacterial suspension equivalent to the 0.5 MacFarland turbidity standard (~1.5 × 10^8^ cfu/mL) was prepared in TSB broth. The inoculum was further diluted in broth to give a final organism density of 1.5 × 10^6^ cfu/mL. Free and liposomal MOX stock solutions of 147 μM each were used for the study. The stock solutions were diluted following a series of two-fold dilutions in sterile TSB broth to range from 0.078 nM to 73.48 μM, and solutions were placed in microtiter wells. A volume of bacterial suspension (100 μL) equal to the volume of the diluted MOX solution (100 μL) was added to each well of antimicrobial agent and incubated at 37 °C for 18 h. The MIC was defined as the lowest concentration of MOX at which no obvious growth was observed. All the experiments were performed in triplicate.

#### 2.3.3. Bacterial Growth Curve Assay

Bacterial Growth rates were spectrophotometrically monitored [30,33] both in the presence and absence of different liposomal or drug solutions (MOX liposomes, empty liposomes, mixtures of free MOX, and empty liposomes or free MOX) at the MIC50 concentration of free MOX (0.15 μM). Briefly, overnight grown bacterial cells from a TSA agar plate were allowed to grow in fresh TSB broth (without glucose) to their early exponential phase. The broth containing bacteria was inoculated into 96-well flat-bottomed polystyrene plates with initial absorbance at λmax 570 nm ~0.01. The change in absorbance of each well was monitored each hour for a total of 24 h by a Fluostar (BMG LABTECH) microplate reader.

For the determination of time–kill curves, a fresh culture of *S. epidermidis* in a final inoculum of ~4 × 10^7^ cfu/mL (in TSB broth) was incubated at 37 °C for 2, 4, 6, and 24 h with 0.15 μM of free or liposomal moxifloxacin (MIC 50 of free moxifloxacin). A culture of the same strain without antibiotics was used as positive control. At each time interval, bacteria were harvested in serial dilutions, and the cfu/mL was determined by counting the single colonies that emerged in the appropriate dilution on TSA plates. Experiments were performed in triplicate.

#### 2.3.4. Biofilm Susceptibility Assays

The antibiofilm activity was evaluated at concentrations corresponding to the MIC of MOX, i.e., 0.3 μM, for all the formulations, according to previously reported methods [31,32,33]. Crystal violet (CV) staining and a validated MTT [3-(4,5-dimethylthiazol-2-yl)-2,5-diphenyltetrazolium bromide, a yellow tetrazole] cell viability assays were used to assess biofilm susceptibility to each free or liposomal MOX formulation. Briefly, one single bacteria colony isolated from fresh agar plates was inoculated into a tube filled with 5 mL sterile TSB and incubated at 37 °C for 24 h. Fresh bacterial suspensions were prepared in TSB with 1% glucose from overnight cultures and adjusted to 0.5 MacFarland turbidity standard, followed by 1:10 dilution into fresh media. Then, 200 μL of the suspension was added to 96 well sterile polystyrene plates and incubated at 37 °C for 24 h. Following overnight incubation, plates were gently washed with 1x phosphate-buffered saline (PBS; pH 7.4) to remove planktonic cells, and the well-formed biofilm was incubated with free or liposomal moxifloxacin at 37 °C for 24 h. The next day the bacterial suspension of each well was gently spent and washed three times with phosphate buffer saline (pH 7.2) and stained with 195 μL of 0.1% Crystal Violet (Sigma-Aldrich, St. Louis, MO, USA) for 15 min at room temperature. Excess crystal violet was removed by washing with tap water, and biofilm was quantified by measuring the corresponding OD-570 nm of the supernatant following the solubilization of CV in 95% ethanol. For each sample (free or liposomal) tested, biofilm assays were performed in triplicate, and the mean biofilm absorbance value was determined.

In the MTT assay, biofilms were incubated with MTT (0.5 mg/mL) at 37 °C for 1 h. After washing, the purple formazan crystals that formed inside the bacterial cells were dissolved by acidified isopropanol and then measured using a microplate reader by setting the detecting and reference wavelengths at 570 nm and 630 nm, respectively.

### 2.4. Statistical Analysis

IBM SPSS statistics pack was used for the statistical analysis of the results. All experiments were performed in triplicate. All data are presented as the mean ± standard deviation of the mean of independent experiments. Statistical significance was evaluated by one-way ANOVA or two-way ANOVA and LSD’s post hoc test with a significance level of *p* < 0.05.

## 3. Results and Discussion

### 3.1. Physicochemical Properties of MOX Liposomes

#### 3.1.1. Effect of Liposome Preparation Method on Moxifloxacin Encapsulation

As shown in Table 1, where physicochemical properties of MOX liposomes are presented, the AL method produced liposomes with 50–81% higher MOX-loading efficiencies as compared to the corresponding (with the same lipid composition) DRV liposomes in all cases. It should be noted at this point that PC liposomes prepared by conventional thin-film hydration method (using the same initial lipid and drug concentrations, as used for the liposomes prepared by DRV and AL) had a MOX encapsulation efficiency equal to 0.35%, more than 30 times lower than the corresponding DRV liposomes, justifying the selection of the DRV method as a passive drug loading method that could be compared with the active loading method.

For the AL method, the ability of all these liposomal types to encapsulate MOX was studied at different pH values (5.0, 5.6, and 6.0) and temperatures (37 °C, 41 °C, and 60 °C) (Appendix A). Concerning pH in interior/exterior liposomal media, optimal loading was realized by loading MOX in liposomes having (NH_4_)_2_SO_4_ pH 5.60 inside and PBS pH 7.40 outside. In all vesicles loaded with MOX at different temperatures/durations, the encapsulation of MOX was practically the same, indicating that temperature or duration of incubation modulation (in the ranges studied) does not alter the loading of MOX.

The effect of the liposome preparation method on MOX encapsulation is better shown in Figure 1, where the encapsulation of MOX in DRV and AL liposomes (expressed as drug/lipid molar ratio, D/L) is shown for all the liposome compositions used.

Accordingly, MOX loading is influenced by the liposome preparation method for all the liposome compositions used; significantly higher encapsulation of MOX was found in AL liposomes as compared to corresponding DRV liposomes (Figure 2A–D), despite the fact that the DRV liposomes were larger compared to the corresponding (with same lipid composition) AL liposomes. Active loading of MOX in liposomes was also performed by Le-Daygen et al., and it was found to confer higher encapsulation compared to thin-film hydration methods (which are considered passive loading methods) [26]. However, we cannot directly compare our results on liposome physicochemical properties and encapsulation efficiencies with the ones reported before [26], since different lipid compositions were used, and loadings were performed under different conditions (initial drug and lipid concentrations), parameters which are known to modulate drug trapping efficiencies, as in the case of remote loading of ciprofloxacin that was affected by initial drug/lipid ratio and lipid composition [34,35].

The highest encapsulation of MOX in AL liposomes was demonstrated in the cases of highly charged (containing 20 mol% PG) (Figure 2A) and PEGylated liposome types (Figure 2D). This can be explained by considering that in (both of) the later cases, vesicles have a lower tendency to come in close contact or aggregate (compared to liposomes that are either uncharged or non-PEGylated) due to repulsive forces between the negatively charged vesicles and steric effects between PEGylated vesicles. Consequently, the total liposomal surface area available for drug/membrane contact and subsequent for drug permeation through the lipid membrane is higher (compared to the corresponding surface area of uncharged or non-PEGylated liposomes), resulting thus in higher amounts of drug being entrapped in the specific liposome types. Nevertheless, the effect of adding PG in liposome lipid membranes on MOX loading is also due to direct interactions between lipids and a positively charged MOX group, as explained below [26].

#### 3.1.2. Effect of Liposome Membrane Lipid Composition on MOX Encapsulation

Concerning the effect of liposomal membrane composition on MOX encapsulation, as seen in Figure 1, the increase of liposome membrane Chol content from 33 mol% to 50 mol% (of total lipid) confers a significant increase in MOX encapsulation (Figure 2B–D) in liposomes prepared by the AL method. Oppositely, the increase of Chol content did not (significantly) modulate MOX encapsulation efficiencies in the cases of liposomes prepared by the DRV method.

The effect of adding PEG, PG, or both, in liposome membranes, on drug encapsulation is better seen in Figure 2. PEG addition in PC and PC/Chol (2:1) liposomes prepared by the DRV method resulted in a significant decrease of MOX encapsulation (Figure 2A).

The latter result can be explained by a “cryoprotectant-like” effect of PEG that may restrict the complete disruption of empty SUV liposomes during the freeze-drying step of DRV formation, resulting in lower amounts of lipid membranes being available for re-structuring of liposomes and concurrent encapsulation of drug during the hydration step [23,24,36]. On the contrary, PEG coating of vesicles prepared by the AL method results in a significant increase of MOX encapsulation in vesicles (in all lipid compositions tested) (Figure 2B), as recently reported also for physostigmine liposomes [37].

Finally, the effect of adding PG in the liposome membrane on liposome MOX loading can be observed in Figure 2C,D. The addition of 20 mol% PG in the vesicle lipid membranes confers a significant increase of MOX encapsulation in all liposome types (lipid compositions) prepared by DRV (Figure 2C) and AL (Figure 2D) methods. In a recent report [26], it was demonstrated that MOX can interact with phosphate and carbonyl groups of DPPC/cardiolipin anionic liposome membranes via its positively charged pyrazole cycle. Similar interactions with the current anionic PG-containing liposomes may explain increased encapsulation of MOX in these liposomes.

It is well-known that the composition of liposome lipid membranes determines lipophilic drug partitioning and encapsulation in liposomes since these drugs are located in the lipid membrane. However, the lipid composition of the liposome membrane additionally affects its fluidity, polarity, and thickness, and thus the encapsulation of hydrophilic drugs is usually affected as well [38,39]. Furthermore, Chol incorporation into liposomal bilayers has an important contribution to membrane organization, dynamics, and function. Chol reduces the phospholipid hydrocarbon chain rotational freedom, thus decreasing the hydrophilic material leakage, while Chol molecules stabilize lipid bilayers increasing drug retention. Additionally, Chol addition in lipid membranes results in increased liposome size and liposomal aqueous core volume (in the majority of cases), thus increasing encapsulation of hydrophilic molecules. Nevertheless, other behaviors have been reported and have been explained on the basis of particular interactions between Chol and specific drugs and/or other lipid components of the particular formulations [39]. The effect of lipid membrane composition on drug encapsulation may also be determined by the particular method used for drug loading, as already mentioned above for the DRV method; the use of very rigid/stable SUV liposomes that are not fully disrupted during the freezing step of DRV formation results in lower drug encapsulation (compared with the use of less rigid SUVs) [24,36].

Concluding, the lipid membrane composition and mostly the addition of PEG and/or PG (at 20 mol% content) confer significant effects on MOX-loading efficiency, especially in liposomes prepared by the AL method.

#### 3.1.3. Effect of Liposome Preparation Method and Lipid Membrane Composition on Vesicle Size Distribution, Zeta Potential, and Morphology

Considering the size distribution of MOX-loaded liposomes prepared by different methods, as observed in Table 1, both preparation methods result in the formation of liposomes with sizes within the nanoscale. All DRV liposomes had mean diameters ranging between 121 and 155 nm, with narrow size distribution (PDI values were between 0.064 and 0.175), as anticipated since they were extruded through 100 nm pore membranes. The liposomes prepared by the AL method had significantly lower mean diameters (compared to corresponding DRV liposomes) between 84 and 105 nm, while their PDI values were higher compared to those of the extruded DRV vesicles ranging between 0.137 and 0.203, as they were initially formulated by sonication. In all cases, the addition or increase of the Chol content of liposomes confers a slight increase in the liposome mean diameter.

Zeta-potential values of vesicles produced by the two methods are determined by their lipid composition; as anticipated, liposomes composed of zwitterionic lipids have close to zero zeta-potential values, while the addition of PG in the lipid membranes of vesicles resulted in negative zeta-potential values that were further increased when PG-content increased from 10 to 20 mol%. When the PG-containing liposomes were PEGylated, decreases in the vesicle zeta potential values were demonstrated, confirming the incorporation of the PEG lipids in the lipid membrane, as reported before [40].

The morphology of the two liposome categories was studied by cryo-TEM. Representative micrographs of DRV and AL liposomes, with PC/Chol (2:1) lipid composition, PEGylated and non-PEGylated, are presented in Figure 3.

As seen, all samples contain liposomes with round morphology, and the vesicle diameters observed verify the corresponding DLS measurements (Table 1). In most of the vesicles observed in AL liposome micrographs, especially the PEGylated ones, a granularity is clearly observed in the aqueous core as if they are filled with the drug. In respect to the differences between the two liposome types, the granularity (higher electron density) is observed in the aqueous core of about half of the non-PEGylated AL liposomes (upper frames in Figure 3) and almost all of the PEGylated AL liposomes (middle frames); however, much fewer DRV liposomes have a similar granular texture. If we assume that the granularity is proportional to the amount of drug encapsulated (and that most of the drug is entrapped in the aqueous core), the observations from the micrographs are in good agreement with the lower drug-loading efficiencies of DRV compared to AL liposomes, and the increased loading reported in the case of PEGylated AL liposomes (compared to the corresponding non-PEGylated liposomes).

If the granularity observed in cryo-TEM micrographs is due to the presence of the precipitated drug in the liposome aqueous core, we may postulate that it is in a non-crystalline phase since it appears very different from observations of doxorubicin-loaded liposomes in previous studies [41]. Nevertheless, other morphologies have been observed for liposomes entrapping precipitated water-soluble acridine and phenanthridine [42], and it was recently reported that the physical state of doxorubicin complexes at the intra-liposomal aqueous phase could be either crystalline or in a “liquid-crystal like phase” with differing cryo-TEM morphologies, according to the type of ammonium counter anion used for establishment of the gradient applied for drug loading [43]. Thereby, we cannot be sure about the state of the encapsulated MOX in the different types of liposomes prepared herein.

XRD spectra were recorded in order to detect potential crystalline structures in the same MOX-loaded liposome types that were observed by cryo-TEM.

As seen in Figure 4, some crystallinity is detected in the AL liposomes, especially in the PC/Chol/PEG liposomes, while no crystallinity is seen in the DRV liposomes. This finding is in good agreement with (i) the amounts of MOX encapsulated in the different types of liposomes (Table 1); (ii) the estimations for liposome entrapped volumes according to a previously published nomograph (relating theoretical captured volume, diameter, number, area, and lipid weight of unilamellar liposomes) [44], and (iii) the solubility of MOX in the entrapped aqueous media of the different liposome types, PBS and (NH_4_)_2_SO_4_, at 37 °C that was experimentally found to be 19.9 ± 1.0 mg/mL and 40.9 ± 3.1 mg/mL, respectively. In more detail, according to the previously reported nomograph [44], for a lipid amount of 5 mg (6.5 μmol) and liposomes with a mean diameter ~100 nm, the entrapped aqueous volume is ~18 μL. Thus, it is calculated that the intraliposomal MOX concentration is approximately 19 mg/mL in the DRV liposomes (practically equal to the solubility of MOX in PBS) and about 35 mg/mL (lower but close to MOX solubility in (NH_4_)_2_SO_4_) and 75 mg/mL (much higher than MOX solubility in (NH_4_)_2_SO_4_) for the non-PEGylated and PEGylated AL liposomes, respectively. Thereby, the later estimations are in agreement with the morphological observations and the XRD results, suggesting that some of the encapsulated MOX in the PC/Chol/PEG AL liposomes were precipitated in the form of crystals inside the aqueous liposomal core. This finding may be connected with the slower release of MOX from the later liposome type, as presented below. In any case, we cannot exclude the possibility that a larger fraction of liposome-encapsulated MOX is entrapped in the lipid membrane of the DRV liposomes, re-compared with AL liposomes where most of the entrapped MOX is most probably retained in the ammonium sulfate containing aqueous core (where it is ionized).

### 3.2. Effect of Preparation Method on Release Kinetics of MOX from Liposomes

The release kinetics of MOX from the two liposome types is reported in Figure 5, where the release of MOX from liposomes prepared by the DRV method (Figure 5A) and AL method (Figure 5B) is shown as a function of time during incubation of the liposomes at 37 °C. As seen, MOX release is significantly faster from liposomes prepared by the DRV method as compared to those (with the same lipid composition) prepared by AL (2-way-Anova, *p* < 0.0001). Indeed, all liposomes prepared by the DRV method release their entire MOX content after 24 h, while at the same time point, the highest release from the AL liposomes is ~65%. In the later liposome type, the lipid composition of liposomes determines the release rate of MOX; liposomes with 50 mol% Chol in their membranes (lipid/Chol 1:1) are the ones that retain higher amounts of MOX for longer time periods. In addition to the increase of the Chol content of liposomes, also coating with PEG is demonstrated to increase MOX retention in liposomes when the liposomes have low Chol content (PC/Chol 2:1) (Figure 5B). As mentioned above, this finding may be explained by the precipitation of MOX in the aqueous core of the liposomes.

The different release kinetics of MOX from the two liposome types is probably associated with the higher degree of MOX ionization in the interior of the liposomes prepared by the AL method that have (NH_4_)_2_SO_4_ inside the liposomes, compared to the ionization of MOX in PBS (in the DRV liposomes). Indeed, higher retention of remotely loaded drugs in liposome has also been demonstrated by others [41], while the increase of Chol content and PEGylation is known to increase the integrity of liposomes in general and thus enhance the retention of drugs in the aqueous core of the vesicles [35,36,39,40,45]. Furthermore, it was reported that the phase of the doxorubicin complex entrapped in AL liposomes affects the drug release profile [43].

In a previous study, 1,2-dipalmitoyl-sn-glycero-3-phosphocholine (DPPC) and cardiolipin (CL2) were demonstrated to form stable complexes with MOX due to the electrostatic interaction of negatively charged nitrogen in the heterocycle moiety of MOX with cardiolipin phosphate groups [26]. DSC studies revealed that in DPPC/CL2 liposomes, MOX formed two types of complexes with the lipids (having different structure and phase transition temperatures) that stabilized the lipid bilayer, but in neutral DPPC liposomes, the interaction of MOX with the liposomes bilayer results in the formation of defects and fluidization of the lipid membrane, and rapid release of MOX [26]. Perhaps this is why MOX is rapidly released from the current DRV liposomes. However, we did not observe any difference in the release kinetics of MOX between neutral and anionic DRV liposomes in the current study (Figure 5A), revealing that perhaps the previously reported interactions between MOX and anionic liposomes (that result in increased liposome integrity) are specific for cardiolipin-containing liposomes.

### 3.3. Physical Stability of MOX Liposomes

The physical stability of the MOX liposomes with the highest drug encapsulation and retention (AL method liposomes) was studied during storage for a period of 90 d at 4 °C. Vesicle size distribution (mean hydrodynamic parameter and polydispersity index [PDI]) and zeta potential were measured. The physicochemical properties of the liposomes at each time point are shown in Appendix A. All liposomes evaluated demonstrate high physical stability during storage at 4 °C, at least for up to 3 months (90 d).

The retention of MOX-loaded PC/Chol (1:1 mol/mol) DRV and AL liposomes was additionally evaluated under the same storage conditions (4 °C for up to 90 d). As seen from the results (Appendix A), MOX is released rapidly from the DRV liposomes, even in this study where incubation was at 4 °C. Oppositely, AL liposomes retain the full amount of MOX even for 120 d when stored at lipid concentrations ≥ 8 mg/mL; when stored at a lipid concentration of 2 mg lipid/mL, some of the MOX (33% of initial amount) is released after 120 d, but even at such low lipid concentrations AL liposomes retain their full drug load for 90 d. These results confirm the high physical stability of MOX-(actively)-loaded (AL) liposomes.

### 3.4. Effect of Liposome Preparation Method on Antimicrobial Activity

It has never been studied up to date how the preparation method of liposomal drugs may affect their antimicrobial activity. Herein we evaluated the antimicrobial activity of MOX-loaded PC/Chol (1:1) liposomes prepared by (i) the DRV method and (ii) the AL method, towards the ATCC 35,984 reference strain of *S.epidermidis*, under identical experimental conditions. It is well known that coagulase-negative staphylococci (including *S. epidermidis*) are responsible for 40–80% of acute post-cataract surgery endophthalmitis, followed by *Staphylococcus aureus*, and to a lesser extent (0–50%) for post-injection and post-traumatic endophthalmitis [46]. For this, we formulated DRV and AL liposomes to have the same D/L ratio by adjusting the amount of MOX incubated with empty liposomes during the active loading procedure for AL liposome preparation. The physicochemical properties of the two types of liposomes that were used in these studies are shown in Table 2.

#### 3.4.1. Inhibition of Planktonic Bacterial Growth and Time–Kill Curves

Before carrying out the experiments of bacterial growth and bacterial killing activity, the MIC-50 of MOX towards planktonic reference strain ATCC 35,984 of *S. epidermidis* was verified by the broth microdilution assay as described in Materials and Methods. As concluded from the results of the previous experiment presented in Appendix A, the previously reported MOX minimum effective concentration-50 (MIC-50) against *S. epidermidis* reference strain of 0.15 μM [29] was verified. In particular, MIC of 0.2451 ± 0.0044 μM and the MIC-50 of 0.1508 ± 0.0045 μM were calculated graphically (Appendix A), whereas the corresponding values calculated by the logistic model fitting of experimental values were 0.2328 μM and 0.1466 μM, respectively (Appendix A). In the same experiment, the superior planktonic antimicrobial activity of liposomal MOX compared to free MOX was demonstrated since the corresponding MIC and MIC-50 for AL MOX-loaded liposomes were found to be 0.09156 μM and 0.05768 μM, respectively.

As shown in Figure 6, the growth of *S. epidermidis* bacteria was substantially inhibited by both types of MOX liposomes tested (DRV_MOX and AL_MOX) (Figure 6A); both liposome formulations could inhibit bacterial growth significantly better as compared to the same concentration of free drug (Free_MOX), which also demonstrated a significant reduction (close to 50%) of bacterial growth over the period evaluated. On the other hand, empty liposomes (with the same lipid composition and at the same lipid concentration (21.5 μM) did not have any significant effect on bacterial growth, whereas the mixture of free drug (0.15 μM) and empty liposomes demonstrated a slightly lower effect on bacterial growth as compared to that of the free drug. The latter result may be explained if we assume that the empty liposomes may be used by bacteria as nutrients, which is in agreement with the slightly higher bacterial growth demonstrated in the presence of empty liposomes (after 20 h of co-incubation) as compared to the control (plain planktonic bacteria).

Interestingly, the DRV_MOX liposomes had a significantly lower inhibitory effect on bacterial growth as compared to that of AL_MOX liposomes (*p* < 0.0001) after about 18 h of co-incubation of liposomes with bacteria (Figure 6A), when most of MOX has leaked out from the DRV liposomes, as observed in Figure 5A. Indeed, at 24 h, the DRV_MOX liposomes conferred close to 70% inhibition of planktonic bacterial growth as compared to more than 90% inhibition by the AL_MOX liposomes (Figure 6A).

An even more pronounced effect of the preparation method (of MOX liposome) on the planktonic antibacterial activity of the liposomes was observed in the bacterial killing study, as seen in the corresponding time–kill curves of Figure 6B. Both liposome types demonstrated significantly higher bacterial killing activity compared to the free drug, while the higher activity of AL_MOX liposomes compared to DRV_MOX liposomes was already demonstrated at 2 h. In fact, the higher killing activity of the DRV_MOX liposomes as compared to the free drug was observed to decline after 6 h, and at 24 h, the DRV liposomes had almost similar killing activity as that of the free drug (Figure 6B,C).

It is interesting that the latter time frame of the killing activity of DRV_MOX liposomes is in agreement with the release kinetics of MOX from the specific liposome type; indeed, 83% of liposome-encapsulated MOX is released from DRV_MOX liposomes after 6 h incubation at 37 °C (Figure 5A).

In a previous study, gentamicin-loaded DRV liposomes (with diameters between 625 to 806.6 nm) were found to inhibit the growth and kill biofilm-forming *Pseudomonas aeruginosa* and *Klebsiella oxytoca* significantly more than the free drug. Neutral liposomes were found to lower the MIC and minimum bactericidal concentration (MBC) of gentamicin against planktonic bacteria, while negatively charged liposomes demonstrated a similar bacteriostatic effect with free gentamicin. Liposomes also inhibited biofilm formation better than free gentamicin. Interestingly, the DRV liposomes were reported to retain >60% of their gentamicin content for the full time period of experiments, backing up our suggestion about the important role of increased drug retention in AL liposomes on their superior activity [47].

Other studies have also reported enhanced activity of liposomal drugs (compared to free drugs) towards planktonic bacteria. In one case, curcumin antibacterial properties were augmented by encapsulation in positive liposomes, and the importance of physicochemical properties of liposomes (bilayer rigidity and liposome charge) as crucial parameters that determine their interaction with the biological environment was pointed out [48]. In another report, the antibacterial properties of cephalexin-loaded niosomes (a type of liposomes that includes surfactants in their membrane) against *S. aureus* (Gram-positive) and *Escherichia coli* (Gram-negative) bacteria were augmented (MIC was reduced by two or four times) compared to the efficiency of the free drug [30]. Nevertheless, in other cases, liposomal encapsulation resulted in decreased antimicrobial properties, as in the case of liposomes loaded with phenylpropenes (PPs) (eugenol, isoeugenol, estragole, and trans-anethole) that were assessed against *E. coli* and *S. epidermidis*. The decreased activity was then explained by prolonged retention of PPs in the liposomes that prevented their interaction with bacteria [49].

To summarize, although the effect of liposome membrane lipid composition on liposome antimicrobial activity towards planktonic bacteria has been studied in several cases, the method used for drug loading and/or liposome preparation has not been considered before, to the best of our knowledge.

#### 3.4.2. Antibiofilm Activity of MOX Liposomes-Effect of Liposomes Preparation Method

The effect of MOX liposome type on the anti-biofilm activity was also studied by performing bacterial biofilm susceptibility assays with the same bacterial strain and the same liposomal preparations, using a MOX concentration of 0.3 μM (equal to MIC) and a lipid concentration of 43 μM. As seen in Figure 7, in established biofilms, the anti-biofilm effect of free MOX at MIC is minimal, in accordance with previous reports [50,51]. Interestingly, AL_MOX liposomes conferred substantial inhibition of biomass (Figure 7A) and significant reduction of the biofilm viability (Figure 7B) as compared to DRV_MOX liposomes (*p* < 0.0001) as well as to free-MOX (*p* < 0.0001). In fact, DRV_MOX liposomes did not significantly reduce the biomass or the viability of bacteria in the biofilm. Thereby, the liposome preparation method is shown to significantly influence the activity of MOX liposomes towards bacterial biofilm, in addition to bacterial growth and killing.

As commented above for liposome formulation activity towards planktonic bacteria, the enhanced anti-biofilm activity of the AL liposomes may also be connected with the higher retention of MOX in the AL liposomes (compared to the DRV liposomes), indicating that the interaction of bacteria with intact vesicles (or vesicles that retain MOX) is an important prerequisite for the expression of bacterial biofilm susceptibility, as well.

In a previous study, although negatively charged tobramycin liposomes displayed increased anti-biofilm effect against *Burkholderia cepacia* complex (Bcc) bacteria (compared to the free drug), the lack of a substantial anti-biofilm effect of neutral tobramycin liposomes was attributed to the low encapsulation efficiency of tobramycin in neutral liposomes, in agreement with our current suggestion [52].

Several other reports have also proved in vitro and in vivo (in some cases) the superior anti-biofilm activity of liposomal drugs. It was indeed proposed that the reduced penetration of antibiotics through biofilm and low accumulation at infected sites can be overcome by nanotechnological platform approaches [53]. In this context, cationic liposomes were found to have a high anti-biofilm effect and prolonged retention on P. aeruginosa biofilm, and anionic liposomes showed high permeability in the biofilms. PEG coating increased the anti-biofilm effect but reduced liposome retention on biofilms. It was concluded that surface charge and PEG modification contribute to the effectiveness of liposomes [54]. However, although positively charged RFB-liposomes demonstrated the highest interaction with methicillin-susceptible *S. aureus* biofilms, the non-cytotoxic negatively charged RFB liposomes were proposed as a preferred liposome type against *S. aureus* infections (due to the cytotoxicity of the positively charged vesicles towards osteoblasts and fibroblasts) [53]. In another case, encapsulation in liposomes augmented the synergistic activity of berberine and curcumin by enhancing phytochemicals uptake in bacterial cells. The liposomes effectively reduced methicillin-resistant *S. aureus* (MRSA) biofilm and associated intracellular infection [32]. The interactions between cationic liposomes and *S. epidermidis* biofilms were shown to be dependent on the ionic strength of the surrounding medium, suggesting that adsorption is mediated by electrostatic effects. Studies using cells with different hydrophobicity proved that hydrophobic effects also play a role in liposome adsorption to biofilms [55]. Elsewhere, to determine what bacterial structures or envelope components may affect bacterial fusion with different liposome types, interactions between different *P. aeruginosa* strains and cationic liposomes were tested, and it was suggested that even one bacterial wall protein may favor stronger interactions between *P. aeruginosa* cells and cationic liposomal formulations [56].

In summary, for the activity of liposomal drugs towards planktonic bacteria, although numerous studies have been conducted to evaluate the effect of different liposomal properties on their anti-biofilm activity, the effect of the liposome preparation method has not been previously evaluated.

### 3.5. Comparison of Current Results with Relevant Literature

When searching for relevant studies in the literature, we found that MOX-liposomes were previously studied for their ocular biodistribution post-intracameral injection into the anterior chamber of rabbit eyes and found to confer a satisfactory release profile from aqueous humor [57]. Moreover, a formulation of MOX and Dexamethazone co-loaded liposomes that were embedded in a collagen/gelatin/alginate hydrogel was recently demonstrated to be efficient for the treatment of corneal infection [58]. Non-liposomal MOX nanotechnologies were also studied previously as systems to improve MOX ocular bioavailability and/or to augment its antimicrobial performance against planktonic bacteria and biofilms, such as (i) lipid-polymer nanoparticles modified with hyaluronic-acid (HA-LCS-NPs) that prolonged MOX pre-corneal retention and ocular bioavailability [59], and (ii) MOX-loaded poly(butyl cyanoacrylate) (PBCA) nanoparticles that were found to inhibit intracellular Mycobacterium tuberculosis growth at ten times lower concentration than free MOX [60]. However, the current results cannot be directly compared with other findings since such MOX-loaded liposome types were not evaluated before.

In general, although several studies have been carried out up to date, there is no consensus about the liposome properties that are required for augmented antimicrobial activity of liposomal antibiotics towards planktonic bacterial populations and biofilms. A liposome size between or lower than 100–200 nm has been reported to allow vesicle penetration into infectious biofilms [61]. The fusogenicity of liposomes or else their ability to fuse with bacterial membranes, which is related to the lipid bilayer fluidity, has also been identified as an important factor that influences liposomal antibiotic activity [62]. Loss of a load of liposomes is mentioned as a disadvantage of liposomal antibiotics in the same report [62], but the effect of drug release kinetics from liposomes, as well as that if the liposome preparation method, were never evaluated in the past.

## 4. Conclusions

Herein we report the development of MOX-loaded liposomes as intraocular therapeutics of endophthalmitis. Two different methods, the DRV and AL methodologies, are used for liposome preparation, and the two liposome types are compared for their physicochemical properties, morphology, stability, MOX encapsulation efficiency, and MOX release kinetics. Vesicles with nine different lipid compositions are constructed by both methods in order to evaluate the effect of Chol, negative charge, and vesicle surface coating with PEG on the vesicle physicochemical properties and MOX encapsulation/release in/from vesicles.

The most important differences between the two liposome types according to the experimental results are that (i) MOX encapsulation by AL liposomes is significantly higher as compared to DRV liposomes with the same lipid composition, whereas the effect of lipid composition on the vesicle properties is different between the two liposome types; and (ii) MOX is released rapidly from all DRV liposomes (irrespective of the lipid membrane composition) as compared to the AL liposomes that retain MOX for prolonged time periods, whereas in AL liposomes increased Chol content and PEGylation furthermore delay the release of MOX from the vesicles.

Interestingly, the antimicrobial properties of MOX-loaded PC/Chol (1:1) liposomes (DRV and AL) towards *S. epidermidis* planktonic and biofilm bacteria were demonstrated to be significantly different; the AL liposomes demonstrated increased and prolonged antimicrobial effects, as compared to the DRV liposomes, whereas both liposome types performed better than free MOX, which was in good agreement with other studies [30,32,47,48,52,53,54,55,56].

In summary, the current results demonstrate that the liposome preparation method may be an important parameter that should be considered when designing liposomal antimicrobials. MOX-loaded liposomes prepared by active loading method demonstrate substantial antimicrobial effects towards planktonic and (most important) also biofilm bacteria, most probably due to the stable retention of the drug in the specific liposome type. Further exploitation of MOX-loaded AL liposomes is well justified in order to evaluate the potential effects of the lipid membrane composition on their antimicrobial properties towards the identification of optimal formulations for in vivo evaluation.

## Figures and Tables

**Figure 1 pharmaceutics-14-00370-f001:**
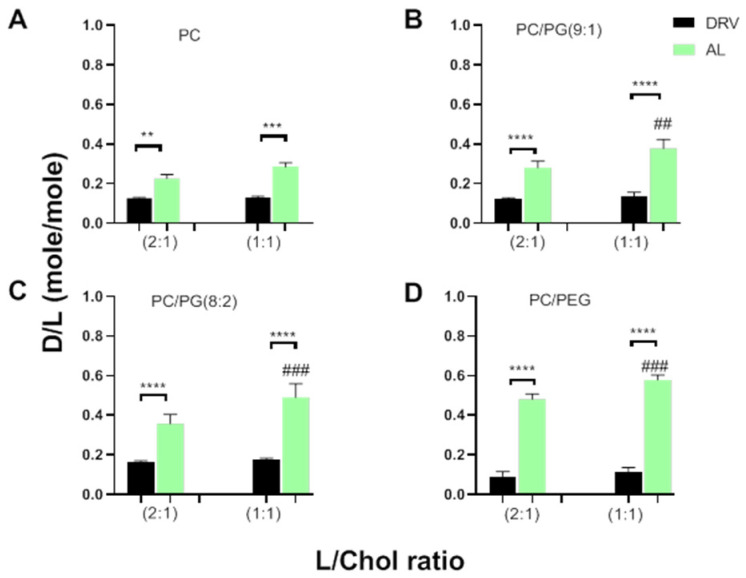
Effect of liposome preparation method (DRV/AL) and Chol content of the lipid membrane on the encapsulation of MOX in liposomes. (**A**) PC/Chol liposomes; (**B**) PC/PG/Chol (9:1:5 or 10) liposomes; (**C**) PC/PG/Chol (8:2:5 or 10) liposomes; (**D**) PC/Chol/PEG liposomes. Encapsulation is expressed as D/L (mol/mol) ratios. Each value reported is the mean of three different samples, while bars represent the corresponding SD of each mean. Statistical significance is denoted by two (*p* ≤ 0.05), three (*p* ≤ 0.001), or four symbols (*p* ≤ 0.0001). Individual differences are marked by * symbols and effect of Chol-content by # symbols (on top of the corresponding bar).

**Figure 2 pharmaceutics-14-00370-f002:**
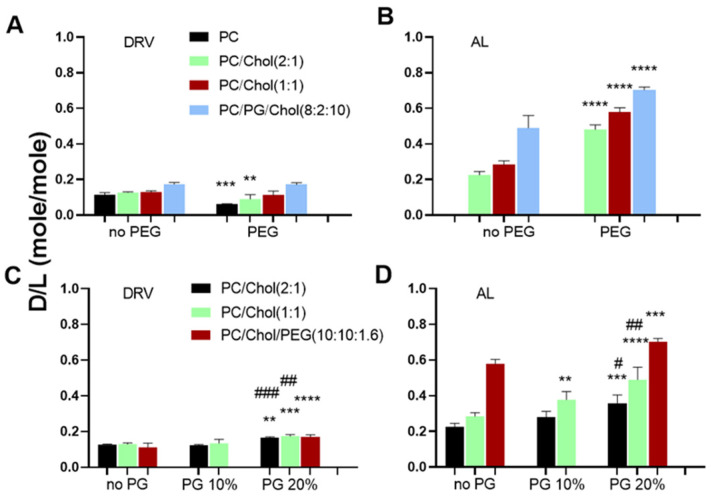
MOX encapsulation (D/L ratios) of liposomes with various lipid compositions; without PEG (no PEG) or with 8 mol% PEG (PEG) in their lipid membrane for DRV Liposomes (**A**) and AL liposomes (**B**), and without PG (no PG) or with 10 mol% PG or 20 mol% PG in their lipid membrane, for DRV liposomes (**C**), and AL liposomes (**D**). Each value reported is the mean of three different samples; bars represent the corresponding SD of each mean. Statistical significance for the effect of addition of PEG or 10 mol% PG in liposomes is denoted by asterisks (*), while the effect of increasing PG content from 10 mol% to 20 mol% is denoted by #; two (*p* ≤ 0.05), three (*p* ≤ 0.001), and four (*p* ≤ 0.0001).

**Figure 3 pharmaceutics-14-00370-f003:**
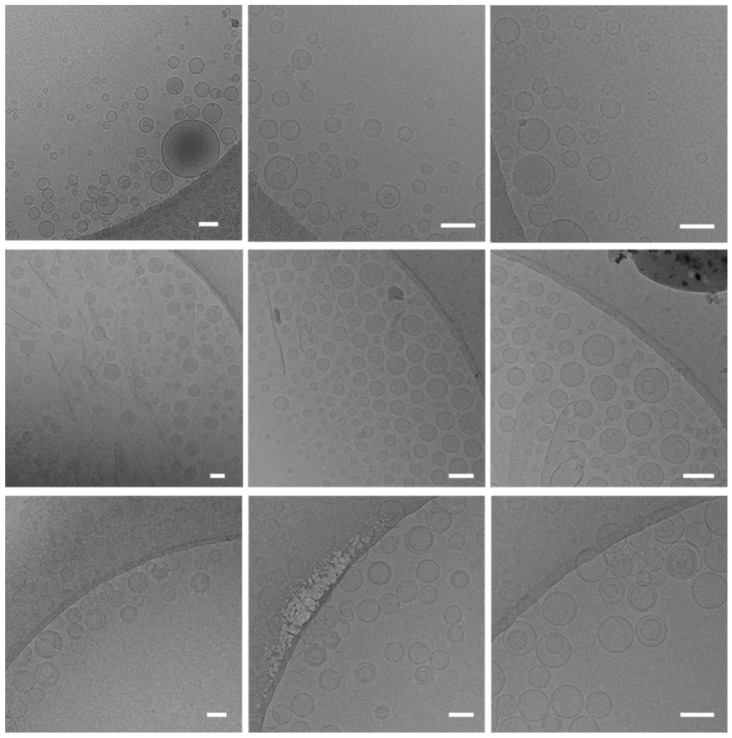
Representative cryo-TEM micrographs of PC/Chol (1:1) liposomes, prepared by AL (upper and middle frames for non-PEGylated and PEGylated liposomes, respectively), and DRV method (lower frames; non-PEGylated liposomes). The size bar corresponds to 100 nm.

**Figure 4 pharmaceutics-14-00370-f004:**
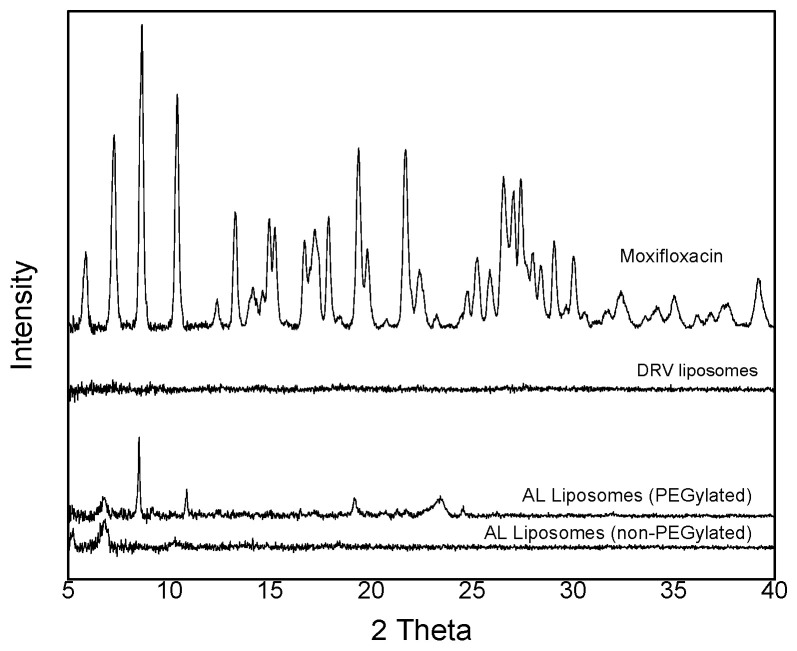
X-ray diffraction spectra of some liposome dispersions (after being precipitated by high-speed centrifugation) as well as solid MOX (for comparison).

**Figure 5 pharmaceutics-14-00370-f005:**
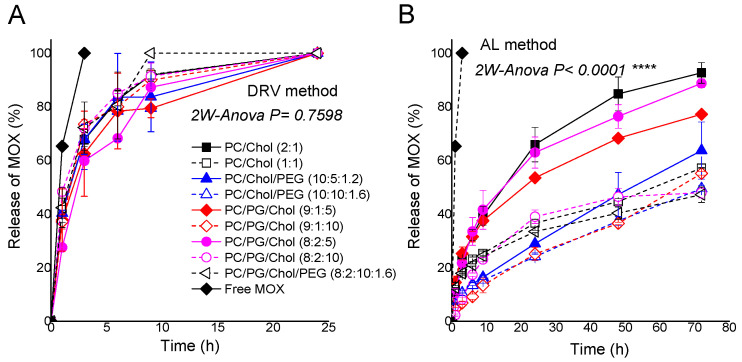
Release of MOX from various types (lipid membrane compositions) of liposomes prepared by DRV method (**A**) and AL method (**B**). Significance of the effect of lipid composition on MOX release kinetics for each method of preparation was studied by ANOVA, and *p* values are reported in the figures. Statistical significance is denoted by four symbols (*p* ≤ 0.0001). Individual differences are marked by **** symbols.

**Figure 6 pharmaceutics-14-00370-f006:**
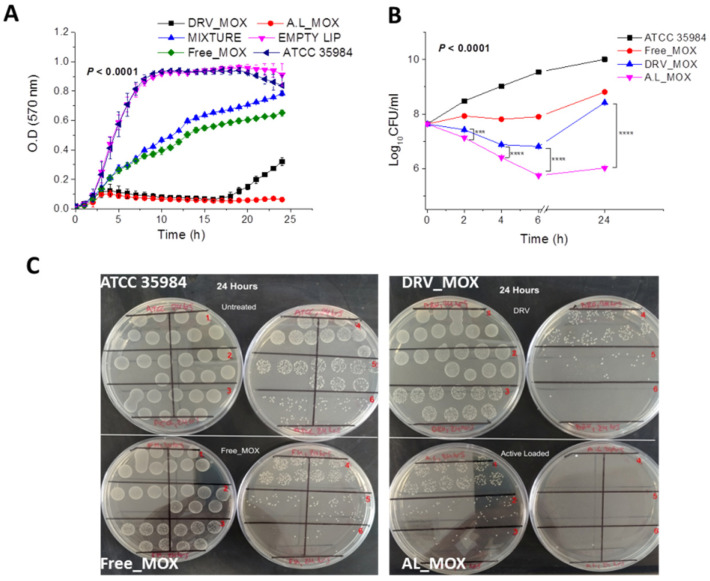
(**A**) Growth curves of ATCC 35,984 *S. epidermidis* bacteria in absence and presence of 0.15 μM MOX as free or liposomal drug (AL and DRV liposomes); empty liposomes and MIXTURE of empty liposomes (21.5 μM lipid) with free MOX were also used as controls; (**B**) time–kill curves of ATCC 35,984 *S. epidermidis* bacteria, in absence and presence of MOX (0.15 μM) as free or liposomal drug (DRV and AL). (**C**) Representative images of single plate-serial dilution spotting with *S. epidermidis* (ATCC 35984) corresponding to 101–106 dilutions after 24 h incubation with free or liposomal MOX from the time-killing studies for each case. Individual differences are marked by * symbols, three (*p* ≤ 0.001), and four (*p* ≤ 0.0001).

**Figure 7 pharmaceutics-14-00370-f007:**
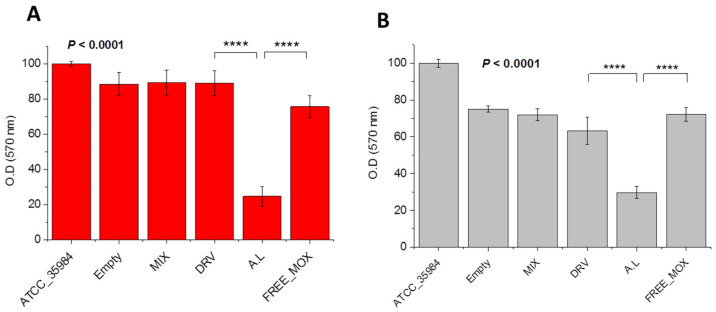
A. Biofilm mass (**A**) and biofilm cell viability (**B**) of ATCC 35,984 *S. epidermidis* bacteria in absence and presence of 0.30 μM MOX as free or liposomal drug (AL and DRV MOX-loaded PC/Chol (1:1) liposomes); Empty liposomes and MIXTURE of empty liposomes (43 μM lipid) with free MOX were also studied as controls. Individual differences are marked by * symbols, four (*p* ≤ 0.0001).

**Table 1 pharmaceutics-14-00370-t001:** MOX encapsulation efficiency (%), mean vesicle diameter, PDI, and zeta potential of various liposomal formulations formulated by the DRV or AL methods. Each value reported is the mean of three different samples and the corresponding SD of each mean.

Lipid Composition	EE (%)	Mean Diameter (Nm)	PDI	Ζ Potential (Mv)
DRV method				
PC	12.1 ± 1.3	121.5 ± 1.9	0.107	−1.24 ± 0.41
PC/Chol (2:1)	13.44 ± 0.29	145.6 ± 6.5	0.116	−3.02 ± 0.28
PC/Chol (1:1)	13.67 ± 0.75	155.2 ± 6.1	0.139	−3.02 ± 0.27
PC/PG/Chol (9:1:5)	12.94 ± 0.59	139 ± 11	0.118	−13.7 ± 1.1
PC/PG/Chol (9:1:10)	14.2 ± 2.3	129.4 ±2.5	0.087	−19.0 ± 2.0
PC/PG/Chol (8:2:5)	17.5 ± 2.2	133.1 ± 1.2	0.089	−22.9 1 ± 0.71
PC/PG/Chol (8:2:10)	18.3 ± 1.1	130.6 ± 1.7	0.064	−23.7 ± 2.1
PC/PEG (10:0.8)	6.41 ± 0.33	115.9 ± 3.6	0.081	−2.85 ± 0.28
PC/Chol/PEG (10:5:1.2)	9.6 ± 2.6	123.3 ± 9.1	0.092	−3.4 ± 1.3
PC/Chol/PEG (10:10:1.6)	11.8 ± 2.4	135.7 ± 7.9	0.081	−3.31 ± 0.75
PC/PG/Chol/PEG (8:2:10:1.6)	18.1 ± 1.1	138.7 ± 8.8	0.175	−5.2 ± 2.1
AL method				
PC/Chol (2:1)	23.9 ± 2.0	84.1 ± 5.3	0.194	−3.2 4± 0.61
PC/Chol (1:1)	30.1 ± 2.2	86.1 ± 2.1	0.179	−2.71 ± 1.1
PC/PG/Chol (9:1:5)	29.6 ± 3.5	84.1 ± 5.3	0.194	−18.1 ± 2.2
PC/PG/Chol (9:1:10)	39.9 ± 4.7	97.6 ± 3.9	0.203	−17.8 ± 1.5
PC/PG/Chol (8:2:5)	37.8 ± 4.9	78.0 ± 4.9	0.200	−28.9 ± 1.8
PC/PG/Chol (8:2:10)	51.8 ± 7.3	89.5 ± 5.1	0.150	−23.4 ± 2.7
PC/Chol/PEG (10:5:1.2)	50.8 ± 2.8	99.5 ± 2.1	0.137	−2.86 ± 0.31
PC/Chol/PEG (10:10:1.6)	61.2 ± 5.6	104.4 ± 4.5	0.141	−2.39 ± 0.21
PC/PG/Chol/PEG (8:2:10:1.6)	74.4 ± 1.7	105.4 ± 4.8	0.154	−3.05 ± 0.18

**Table 2 pharmaceutics-14-00370-t002:** Physicochemical properties of PC/Chol 1:1 (mol/mol) DRV and AL liposomes that were used for in vitro antimicrobial activity studies. The D/L ratio for both liposome types was 0.005.

Liposome Type	Mean Hydr. Diameter (nm)	Polydispersity Index
DRV	119.3 ± 9.1	0.082
AL	104.9 ± 2.1	0.154

## Data Availability

Raw data are available upon request.

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
