# Peer review of "Moxifloxacin Liposomes: Effect of Liposome Preparation Method on Physicochemical Properties and Antimicrobial Activity against Staphylococcus epidermidis"

_pharmaceutics, 2022, doi:10.3390/pharmaceutics14020370_

Round 1

Reviewer 1 Report

Regarding to Manuscript ID: pharmaceutics-1570351
Type of manuscript: Article
Title: Moxifloxacin liposomes: Effect of liposome preparation method on
physicochemical properties and antimicrobial activity against Staphylococcus
epidermidis
Authors: Evangelos Natsaridis, Foteini Gkartziou, Spyridon Mourtas, Marc C.A.
Stuart, Fevronia Kolonitsiou, Pavlos Klepetsanis, Iris Spiliopoulou, Sophia G
Antimisiaris*

The authors prepared and tested different liposomes loaded with moxifloxacin.

Beside the lipid composition of liposome membrane, pH, and temperature, seems that the liposome preparation method is a key factor in design of liposomes which influenced the physicochemical and antimicrobial property of liposomes.

Q1: Why they used S. epidermidis bacteria in this study? Is it a pathogen specific to a disease??

Q2: The studied liposomes are cationic, anionic or neutral ones?

Author Response

Reviewer 1

Comments and Suggestions for Authors

 The authors prepared and tested different liposomes loaded with moxifloxacin. Beside the lipid composition of liposome membrane, pH, and temperature, seems that the liposome preparation method is a key factor in design of liposomes which influenced the physicochemical and antimicrobial property of liposomes.

Answer: We thank this reviewer for agreeing with our main conclusions

Q1: Why they used S. epidermidis bacteria in this study? Is it a pathogen specific to a disease??

Answer: It is well known that coagulase-negative staphylococci (including S. epidermidis) are responsible for 40-80% of acute post-cataract surgery endophthalmitis, followed by Staphylococcus aureus, and to a lesser extend (0-50%) for post-injection and post-traumatic endophthalmitis [Durand ML. 2017. Bacterial and fungal endophthalmitis. Clin Microbiol Rev 30:597–613. https://doi.org/10.1128/CMR.00113-16]. The previous sentence was added in the revised ms, page 14, §3.4, lines 5-8, and the reference was added as REF # 46.

Moreover, the methodologies applied for testing the activity of liposomes in the present study, Bacterial growth curve and Biofilm susceptibility assays (CV and MTT), have already been evaluated and used in previous studies with S. epidermidis strains in our laboratories(see ref 32).

Q2: The studied liposomes are cationic, anionic or neutral ones?

Answer: As evidenced from the zeta potential measurements presented in Table 1 (Page 7) and commented in Paragraph 3.1.3 (Page 10), we studied neutral and anionic liposome (the ones with PG in their lipid membrane) types. Since we did not need to encapsulate any oligonucleotides or substances with high negative charge we avoided to use cationic lipids, due to their toxicity.

Reviewer 2 Report

Manuscript entitled Moxifloxacin liposomes: Effect of liposome preparation method on physicochemical properties and antimicrobial activity against Staphylococcus epidermidis

The authors competently described the characterization and the antimicrobial effect of several liposomal formulations loaded with MOX by different strategies. DRVs and active loaded (AL) formulations were prepared and their properties compared. The properties and efficiency of Als are clearly illustrated and confirms the appropriateness for this intended approach. The major weakness in this manuscript seems to be the DRVs used as a kind of reference, which is in fact also repeatedly argued by the authors in the discussion .

Major comments considering this point are argued below

The applied, very complex procedure preparing the liposomes is not favorable

Line 112, 329 and more - why are the MLVs small sized by sonication and not extruded, although the extrusion technique is milder avoiding lipid damage. A simplified procedure might be applied to create homogenous, more stable MOX-DRV suspensions. Is there any reason applying the two-step method.

Line 118 ….1 mL of the SUV suspension prepared as mentioned above, was 118 mixed with 1 ml of MOX solution in distilled H2O…..Why were prepared SUVs used for MOX loading and the MOX solution not added to the film reconstitution, one step before. Why was distilled water used and for the active loading PBS

Line 404 – MOX granularity and the impact on release and antimicrobial effect – it seems to be obvious, that MOX is differently present in Als and in DRVs. This fact could be the major difference between both formulations which causes the significant difference. This circumstance is not evaluated and therefore hampers the overall comparability, although the authors repeatedly argued that the slower release effects their results.

Chapter 3.1.2 Effect of Liposome membrane lipid composition on MOX encapsulation – the differences of encapsulation efficiencies are extensively discussed, however the results between ALs and DRVs are controverse in respect to PEG, PG and Chol. Thus, if higher encapsulation is achieved by AL in the presence of PEG and PG contradicts with the DRV results. The authors should explain their results in more detail in. Furthermore, as mentioned above it should be explained why this two-step procedure for DRV preparation was used instead of more simple one-step procedure. Finally, an increase of MOX concentration for passive entrapment could be proven.

In conclusion, the manuscript indicates for me that the preparation method determines the properties of the liposomes. This only partially true, even more the entrapped MOX concentration resulting in different morphology determines the characteristics. If minor amounts inside the Als or higher concentration in DRV indicate different behavior is currently speculative. In the light of these findings the authors should rigorously review their manuscript

Minor comments

Line 64 -  intra-ocular instead of intraocular

Line 150 -  Liposomes were analyzed after being lysed in methanol…..how could you assess the efficiency of this method although the phospholipids are poorly soluble in methanol

156 -  of MOX leaded

Line 170 -  50,000 not 50000

Line 264 -  Effect of MF preparation method on MOX EE….MF and EE should be written out

Line 276 -  why are AL preparations significantly smaller and not as indicated slightly smaller as DRVs

Line 300-318 -  is only the repulsive effect responsible or also attraction between MOX and liposome surfaces

Line 399 - …most of the drug is entrapped….is membrane association likely and if yes in which extent

Line 609 – which mechanism causes antimicrobial and biofilm susceptibility? Is MOX released in close proximity to the bacteria or are lipid vesicles adsorbed or absorbed by the bacteria

Author Response

Reviewer 2

Comments and Suggestions for Authors

Manuscript entitled Moxifloxacin liposomes: Effect of liposome preparation method on physicochemical properties and antimicrobial activity against Staphylococcus epidermidis

The authors competently described the characterization and the antimicrobial effect of several liposomal formulations loaded with MOX by different strategies. DRVs and active loaded (AL) formulations were prepared and their properties compared. The properties and efficiency of Als are clearly illustrated and confirms the appropriateness for this intended approach. The major weakness in this manuscript seems to be the DRVs used as a kind of reference, which is in fact also repeatedly argued by the authors in the discussion.

Answer: We would like to thank this author for his/her complete reviewing and suggestions to improve our ms.  We believe that this reviewer did not understand the reason of using the DRV liposomes, as a passive drug loading method into liposomes to compare with the actively loaded liposomes, and show the difference between active loading and passive loading methods. For this reason, we tried to make this point clearer in the revised version of our ms. however we cannot provide a long explanation of DRV method; references are added for this reason, and readers that are not familiar with the method can refer to the specific papers and chapters.

In addition to the revision mentioned below, we added some clarifications in the revised version, and more particular (red parts were added):

-In the Introduction, in the third paragraph a section has been revised to read as  “For this purpose, two different methodologies were evaluated; the Dehydration Rehydration vesicle (DRV) method, which is known to confer high drug loading in liposomes [23,24], and the active loading (AL) method [25, 26], which was previously found to confer higher encapsulation of MOX in liposomes as compared to conventional passive loading or such as  thin film hydration methods [26]. For the later reason, the DRV method was selected as a passive loading method to compare with active loading.”

-In Methods section 2.1.1 (Page 3) the first sentence reads: ”For Dehydration Rehydration vesicle (DRV) preparation, empty small unilamellar vesicles (SUV) were initially prepared as described in detail before [23, 24].”

Also a last line was added that reads: “Extrusion was used as a size-reduction method, in order to prevent disruption of the DRV liposomes and leakage of the encapsulated drug.”

-In Results section  3.1.1 (Page 7) in the first paragraph we added the result of EE for liposomes prepared by the thin-film hydration method , were it reads “It should be noted that PC liposomes prepared by the conventional thin film hydration method (using the same initial lipid and drug concentrations, as used for the liposomes prepared by DRV and AL) had a MOX encapsulation efficiency equal to 0.35%, more than 30 times lower than the corresponding DRV liposomes, justifying the selection of the DRV method as a passive drug loading method that could be compared with the active loading method

 Major comments considering this point are argued below

The applied, very complex procedure preparing the liposomes is not favorable

Line 112, 329 and more - why are the MLVs small sized by sonication and not extruded, although the extrusion technique is milder avoiding lipid damage. A simplified procedure might be applied to create homogenous, more stable MOX-DRV suspensions. Is there any reason applying the two-step method.

Answer:  The DRV method is a mild two step method that results in higher encapsulation of aqueous soluble drugs, by passive loading, compared to other passive loading methods, such as the thin film hydration method. It applies a freeze drying and rehydration step, during which a 10times higher drug concentration is in presence with the lipid membranes during the first part of the rehydration step, resulting in about 10 times higher encapsulation of drugs, compared to the thin film method. At the end the size of DRVs is reduced by extrusion, since if we apply sonication, membranes will be disrupted and the high amount of drug inside the liposomes, will again equilibrate with the free drug concentration outside of the liposomes; thereby all the effort to active high amount of passively loaded drug would be out ruled!

When the DRV method is applied we usually use sonicated SUV liposomes, since this is faster and simpler to produce in labs that have probe sonicators. To use extrusion at this step would be more tedious (since one has to extrude 12-120 times, and use low lipid concentrations, and then concentrate the samples by centrifugation or ultrafiltration, to have the required lipid concentration in the liposome/free drug mixture that will be freeze dried).

Since thin-film rehydration resulted in very low encapsulation of MOX (we have added the result for MOX encapsulation efficiency for PC liposomes prepared by thin-film hydration (0.35%) in section 3.1.1, as mentioned above), we used the DRV method to achieve higher loadings with a passive drug loading method, in order to be able to compare with the AL method.

If we use MLV liposomes in the first (freeze drying) step, the drug encapsulation would be  much lower.

All these points are explained in references 23 and 24 (in the revised version), where the mechanism of drug entrapment by DRV liposomes is explained.

 Line 118 ….1 mL of the SUV suspension prepared as mentioned above, was  mixed with 1 ml of MOX solution in distilled H2O…..Why were prepared SUVs used for MOX loading and the MOX solution not added to the film reconstitution, one step before. Why was distilled water used and for the active loading PBS

Answer:  As explained above, the two methods DRV and AL, are completely different; the one is a passive loading method and the other (AL) an active loading method where the drug is in non-ionized form during loading (this is why PBS is used), and when loaded in the pre-formed SUV liposomes, it becomes ionized and thereby it is trapped in the liposome core, explaining its slow release (compared to the release from DRV liposomes, where the drug is entrapped in its non-ionized –more lipid soluble-form).

DRV liposomes are also prepared in PBS, but due to osmotic pressure difference monitoring, initially 10% PBS is added and this is why the drug is added as a solution in water; elsewise the DRV method would not be resulting in higher drug encapsulation (compared to other passive loading methods); We could not explain all the details of the DRV method in this ms, since this is not the purpose of this research. The authors provide the references, so readers that are not familiar with the method can read and understand the methodology better, since this manuscript is not about the DRV method, but the DRV method is used for the reasons explained above

 Line 404 – MOX granularity and the impact on release and antimicrobial effect – it seems to be obvious, that MOX is differently present in Als and in DRVs. This fact could be the major difference between both formulations which causes the significant difference. This circumstance is not evaluated and therefore hampers the overall comparability, although the authors repeatedly argued that the slower release effects their results.

Answer: Of course this aspect is evaluated, and this the reason why we conducted Cryo-TEM and XRD studies in the different types of liposomes. The release kinetics are probably different due to the different presence of the drug in the two liposome types.

Chapter 3.1.2 Effect of Liposome membrane lipid composition on MOX encapsulation – the differences of encapsulation efficiencies are extensively discussed, however the results between ALs and DRVs are controverse in respect to PEG, PG and Chol. Thus, if higher encapsulation is achieved by AL in the presence of PEG and PG contradicts with the DRV results. The authors should explain their results in more detail in. Furthermore, as mentioned above it should be explained why this two-step procedure for DRV preparation was used instead of more simple one-step procedure. Finally, an increase of MOX concentration for passive entrapment could be proven.

Answer: The differences are explained in detail in the first version of the manuscript, and we do not see any controversy. Please see also the answer below (under minor point  L300-318) for more details.

 In conclusion, the manuscript indicates for me that the preparation method determines the properties of the liposomes. This only partially true, even more the entrapped MOX concentration resulting in different morphology determines the characteristics. If minor amounts inside the Als or higher concentration in DRV indicate different behavior is currently speculative. In the light of these findings the authors should rigorously review their manuscript

Answer: We do not agree with this comment, since the liposomes used in the antimicrobial activity estimation studies had the same MOX concentrations entrapped in the same amounts of lipid, so only the method of drug loading differed. (This is clearly stated in the first paragraph of § “3.4 Effect of Liposome preparation method on antimicrobial activity”, where is reads “It has never been studied up-to-date how the preparation method of liposomal drugs may affect their antimicrobial activity. Herein we evaluated the antimicrobial activity of MOX-loaded PC/Chol (1:1) liposomes prepared by: (i) the DRV method and (ii) the AL method, towards the ATCC 35984 reference strain of S.epidermidis, under identical experimental conditions. For this, we formulated DRV and AL liposomes to have the same D/L ratio, by adjusting the amount of MOX incubated with empty liposomes during the active loading procedure for AL liposome preparation “ )

Thereby, it is indeed the method of drug loading, that affects the kinetics of drug release from the vesicles and maybe also the particular liposome loading space, that determines the antimicrobial efficacy of the liposomes;

Minor comments

Answer: We thank this reviewer for pointing out all the errors, which we corrected in the revised version

Line 64 -  intra-ocular instead of intraocular

Answer: this was corrected

Line 150 -  Liposomes were analyzed after being lysed in methanol…..how could you assess the efficiency of this method although the phospholipids are poorly soluble in methanol

Answer: This is routine method we use in our lab in many research projects. In fact liposomes are fully lysed when diluted x5 in Methanol, here we used 10 times dilution. Phospholipids, at least the ones we usually use are fully soluble in Methanol, and this is the reason why the lipid solutions are initially prepared in the chloroform/methanol mixture (see line 105 in the first version). In fact it is clearly mentioned already in the first version of our ms that the calibration curve was also prepared in the same media with the same procedure. “Liposomes were analyzed after being lysed in methanol; one volume of sample was mixed with 10 volumes of methanol and the mixture was agitated by vortex. A calibration curve in the range 1 -80 μg/mL was constructed by preparation of standard solutions of MOX in media with similar composition as the samples.”

156 -  of MOX leaded

Answer: this was corrected

Line 170 -  50,000 not 50000

Answer: this was corrected

Line 264 -  Effect of MF preparation method on MOX EE….MF and EE should be written out

Answer: This was a mistake; we thank this reviewer for pointing out. It was corrected in the revised version.

Line 276 -  why are AL preparations significantly smaller and not as indicated slightly smaller as DRVs

Answer: We thank this reviewer for pointing out this discrepancy. We agree that the sizes of the two groups of liposomes are significantly different, the DRV liposomes being larger (p<0.05). We corrected this is the revised version, in paragraph §3.1.3, Lines 5-10 , where it now reads: “The liposomes prepared by the AL method had significantly lower mean diameters (compare to DRV liposomes) between 84-105nm, while their PDI values were higher compared to those of the extruded DRV vesicles, ranging between 0.137-0.203, as they were initially formulated by sonication. In all cases, the addition or increase of the Chol content of liposomes confers a slight increase in the liposome mean diameter.” In any case this size difference is not influencing the conclusion, since the larger DRV liposomes encapsulate lower amounts of MOX, despite their larger size (as commented in § 3.1.1,  Page 8, 1st paragraph, revised version).

Line 300-318 -  is only the repulsive effect responsible or also attraction between MOX and liposome surfaces

Answer: For the case of PG, this is not the only explanation. This has been explained below in the following part of the section (in the original version), but in order to connect the different explanations, we added at the end of this paragraph that “Nevertheless, the effect of adding PG in liposome lipid membranes on MOX loading is also due to direct interactions between lipids and a positively charged MOX group, as explained below [26].”

Line 399 - …most of the drug is entrapped….is membrane association likely and if yes in which extent

Answer: We cannot be sure about the extent of MOX association within the membrane; nevertheless, we suppose that some of the drug is probably in the lipid membrane. We agree that this can also be a potential difference between the two liposome types. For this we added a paragraph at the end of section 3.1 on Page 12 of the revised version where it reads as” In any case we cannot exclude the possibility that a larger fraction of liposome-encapsulated MOX is entrapped in the lipid membrane of the DRV liposomes, recompared with AL liposomes where most of the entrapped MOX is most probably retained in the ammonium sulfate containing aqueous core.”

Line 609 – which mechanism causes antimicrobial and biofilm susceptibility? Is MOX released in close proximity to the bacteria or are lipid vesicles adsorbed or absorbed by the bacteria

Answer: According to the difference in release kinetics between DRV and AL liposomes, and due to the substantially higher antimicrobial effect of AL liposomes, compared to the same concentration of MOX in DRV liposomes, we conclude that the mechanism should involve some kind of direct interaction between MOX-retaining-lipid vesicles and bacteria. Whether this leads to adsorption or absorption, and whether MOX is released in close proximity to the bacteria, we cannot know.

Reviewer 3 Report

According to the title - Moxifloxacin liposomes: Effect of liposome preparation method  on physicochemical properties and antimicrobial activity  against Staphylococcus epidermidis- is a  complex and well-argued research about the important role of liposome preparation method  in  the development of Mox-liposomes suitable formulation as intraocular therapeutics of endopthalmitis.

Overall, this is an interesting research suggesting the possibility of a therapeutic application.

The results presented in this paper could be further considered to establish the optimal Mox-liposome formulation and evaluate the antimicrobial  properties in vivo.

The results are in agreement with the objective of this study.

The physico-chemical techniques used for manufacture and charaterisation of  Mox-liposome  formulations  are suitable, well selected and described in detail.

Charts, tables and figures are comprehensive and well explained.

References are suitable for this subject.

I have a suggestion:

This Special Issue is  "A Commemorative Issue in Honor of Professor Gregory Gregoriadis: Liposomes for the Delivery of Drugs and Vaccines".

I suggest to the authors to include  the reference  :

Kirby C, Gregoriadis G (1984) Dehydration-rehydration vesicles: A simple method for high yield drug entrapment in liposomes. Biotechnology 2:979–984,

this being the first work to describe the DRV technique to obtain dehydrated liposome which can be stored for extended periods of time and then rehydrated when and where they are to be used.

Author Response

Reviewer 3

Comments and Suggestions for Authors

According to the title - Moxifloxacin liposomes: Effect of liposome preparation method  on physicochemical properties and antimicrobial activity  against Staphylococcus epidermidis- is a  complex and well-argued research about the important role of liposome preparation method  in  the development of Mox-liposomes suitable formulation as intraocular therapeutics of endopthalmitis.

Overall, this is an interesting research suggesting the possibility of a therapeutic application.

The results presented in this paper could be further considered to establish the optimal Mox-liposome formulation and evaluate the antimicrobial  properties in vivo.

The results are in agreement with the objective of this study.

The physico-chemical techniques used for manufacture and charaterisation of  Mox-liposome  formulations  are suitable, well selected and described in detail.

Charts, tables and figures are comprehensive and well explained.

References are suitable for this subject.

Answer: We are totally grateful to this reviewer for all the positive comments about our paper and research!

 I have a suggestion:

This Special Issue is  "A Commemorative Issue in Honor of Professor Gregory Gregoriadis: Liposomes for the Delivery of Drugs and Vaccines".

I suggest to the authors to include  the reference  :

Kirby C, Gregoriadis G (1984) Dehydration-rehydration vesicles: A simple method for high yield drug entrapment in liposomes. Biotechnology 2:979–984,

this being the first work to describe the DRV technique to obtain dehydrated liposome which can be stored for extended periods of time and then rehydrated when and where they are to be used.

Answer: We could not agree more with this reviewer and this is our mistake. We thank this reviewer for pointing out this omission from our manuscript. We added the reference as ref. 23 in the revised version and re-numbered all the following references

Round 2

Reviewer 2 Report

Dear Authors,

in principle it´s obvious why DRV liposomes were used, however this fact was not figured out in the manuscript. Now it is more precisely depicted and substantiated in your manuscript.

Which method is applied for SUV preparation depends mainly on the established lab techniques and both opportunities are applicable.

Other comments are now answered and more clearly explained in the manuscript